# Emerging Trends in Ant–Pollinator Conflict in Extrafloral Nectary-Bearing Plants

**DOI:** 10.3390/plants13050651

**Published:** 2024-02-27

**Authors:** Eduardo Soares Calixto, Isabela Cristina de Oliveira Pimenta, Denise Lange, Robert J. Marquis, Helena Maura Torezan-Silingardi, Kleber Del-Claro

**Affiliations:** 1Entomology and Nematology Department, University of Florida, Gainesville, FL 32611, USA; calixtos.edu@gmail.com; 2Postgraduation Program in Entomology, Department of Biology, University of São Paulo, Ribeirão Preto 14040-900, SP, Brazil; isalpj@hotmail.com (I.C.d.O.P.); hmtsilingardi@gmail.com (H.M.T.-S.); 3Department of Biology, Federal University of Technology—Parana, Campus Santa Helena, Santa Helena, Curitiba 80230-901, PR, Brazil; deniselange.bio@gmail.com; 4Department of Biology and the Whitney R. Harris World Ecology Center, University of Missouri, St. Louis, MO 63121, USA; robert_marquis@umsl.edu; 5Institute of Biology, Universidade Federal de Uberlândia, Uberlândia 38405-240, MG, Brazil

**Keywords:** protective mutualism, extrafloral nectar, costs and benefits, flower distraction hypothesis, pollination, ant–plant interaction, evolutionary ecology, Cerrado, herbivores

## Abstract

The net outcomes of mutualisms are mediated by the trade-offs between the costs and benefits provided by both partners. Our review proposes the existence of a trade-off in ant protection mutualisms between the benefits generated by the ants’ protection against the attack of herbivores and the losses caused by the disruption of pollination processes, which are commonly not quantified. This trade-off has important implications for understanding the evolution of extrafloral nectaries (EFNs), an adaptation that has repeatedly evolved throughout the flowering plant clade. We propose that the outcome of this trade-off is contingent on the specific traits of the organisms involved. We provide evidence that the protective mutualisms between ants and plants mediated by EFNs have optimal protective ant partners, represented by the optimum point of the balance between positive effects on plant protection and negative effects on pollination process. Our review also provides important details about a potential synergism of EFN functionality; that is, these structures can attract ants to protect against herbivores and/or distract them from flowers so as not to disrupt pollination processes. Finally, we argue that generalizations regarding how ants impact plants should be made with caution since ants’ effects on plants vary with the identity of the ant species in their overall net outcome.

## 1. Ant Protection Mutualisms Mediated by Plant-Based Resources

The nature of interspecific interactions, whether positive or negative, depends on the balance sheet of the contributions of each species to these interactions. In protective mutualisms, ants are attracted to foraging on plants because the plants provide food rewards (Figure 1A,B), domiciles, or both [1,2,3,4,5,6,7,8]. In turn, when acting as mutualists (Figure 1C–G), ants harass, bite, eat, chase off, or otherwise dissuade herbivores from eating the plants [3,9,10,11] (Figure 1H–J). The plants provide a reward to benefit the ants [10,12,13,14,15,16,17], and from the plant’s balance sheet, the cost of the food award or domicile may be outweighed by the benefit of the reduced tissue loss due to reduced herbivory [2,18]. Meta-analyses reveal that, in general, ants visiting plants reduce herbivory and increase seed production, demonstrating a positive balance sheet [19,20]. However, not all ant species are equally effective guards, and some are cheaters, providing no protection at all [21,22].

In protective mutualisms, there is increasing evidence that, in some cases, ants’ interference in the pollination process can be a negative entry or liability on a plant’s balance sheet [21,23,24,25,26,27,28,29]. Ants, otherwise attracted to plants because of their extrafloral nectar, can rob floral nectar, damage floral parts, reduce pollen viability, and chase away potentially effective pollinators. In fact, the “flower distraction hypothesis” was proposed as an explanation for the evolution of extrafloral nectaries (EFNs) when such interference occurs [30,31,32]. This hypothesis states the EFNs evolved to distract ants away from the floral parts to reduce their negative impacts on the pollination process. 

In this review, we focus on studies that demonstrate that ants, while visiting plants with EFNs, bother or chase away floral visitors, including pollinators, in some cases reducing seed set. In their study experimentally testing the flower distraction hypothesis, Wagner and Kay [32] refer to a single study showing that ants influence floral visitors, in this case in a plant species without EFNs. The number of studies on ant–pollinator conflict has increased in the past few years [23,24,25,26,29,30,33], although the outcomes of this relationship in terms of plant fitness remain unresolved. Our specific goals are to (i) review the known negative effects that ants have on pollination and consequently on plant fitness; (ii) describe the factors that might reduce the impact of ants on the pollination process; and (iii) suggest the existence of “optimal ant partners” (and consequently non-optimal ant partners) in the protection mutualism system, similar to the role of effective pollinators in pollination interactions [34]. Importantly, we suggest that optimality of the ant partner not only depends on its contribution to plant fitness via protection but is also balanced against its potential negative effect on plant fitness via pollinator dissuasion. (iv) Finally, we discuss the kinds of plants, specifically in terms of their architectures, phenologies, and floral morphologies, which are most likely to be prone to the negative effects of ants on pollination processes. Overall, this study suggests a broader view of the costs and benefits provided by ants attracted by EFNs, suggesting the existence of optimal patterns which have the highest positive effects on plant protection and the lowest negative effects on pollination processes (Figure 2).

## 2. Negative Effects of Ants on Pollinators

Although ants can protect reproductive parts [35,36,37,38], their presence on flowers may generate negative effects on pollination [31], such as when ants repel or prey on potential pollinators, negatively impacting the reproductive success of the plant [23,24,25,26,27,28]. For instance, in the Neotropics, *Ectatomma tuberculatum* ants occupy certain parts of the plant (Figure 1), such as the EFNs (13 ± 9 s, mean ± SD) and reproductive parts (23 ± 11 s), more often than other ant species, such as *Camponotus crassus* (EFN: 1 ± 1 s, flower: 2 ± 1 s) [25]. When close to the flowers, *E. tuberculatum* will deter or even attack pollinators [25]. In general, few studies have evaluated the direct and indirect impacts that aggressive ants like *Ectatomma* have on pollinators or plant fitness, especially for EFN-bearing plants [24,25,26,27,29,39,40,41,42]. We believe that the identity of the ants and pollinators can be very important to this trade-off [25,27,43]. Ant identity can be understood as a combination of shape, size, aggressiveness, and odor, and pollinator identity can be understood as a combination of size, agility, and likelihood of visiting or not visiting a flower occupied by ants [44]. Therefore, the identity of the ants, as well as that of the floral visitors, can impact the fitness of plants in different ways. 

Studies have shown that very aggressive ants can deter or even prey on pollinators, significantly changing pollinator behavior and/or reducing fruit and seed set [26,27]. Ness [27] observed that the visitation rates and duration of visits to the flowers of *Ferocactus wislizeni* (Cactaceae) by pollinators differed according to the level of aggressiveness of the ants attending the host plants. Flowers occupied by the aggressive ant *Solenopsis xyloni* were visited less frequently and for less time, resulting in fruits with significantly smaller and lighter seeds compared to those of plants with more docile ants. These results show that very aggressive ants can restrict or decrease the reproduction of plants via the disruption of pollination processes in certain environments; however, see [40]. 

Some pollinators are able to identify ant body shape and avoid visiting flowers occupied by ants [24,25,45]. Villamil et al. [26] found that in *Turnera velutina* (Passifloraceae), the presence of ant corpses, experimentally placed in flowers, reduced the duration of visits by pollinators, but these results were contingent on the ant species. The greatest negative effects were caused by the most aggressive ants. In addition, Calixto et al. (in review) studied three species of ants of different sizes and levels of aggressiveness in *Qualea multiflora* (Vochysiaceae) in the Brazilian savanna. They showed that the presence of dead ants in the flowers, again experimentally placed there, decreased plant fitness by preventing the visitation of potential pollinators. They found that the treatment with the largest and most aggressive ants was the treatment most avoided by pollinators. In experiments using plastic ants, Assunção et al. [25] showed that ants were recognized as a danger to pollinators, *Heteropterys pteropetala* (Malpighiaceae) with plastic ants produced fewer fruits than flowers with plastic circles or control flowers. In a similar study, Nogueira et al. [24] showed that pollinators hesitated to visit flowers with artificial ants, negatively affecting pollination, but did not hesitate to visit flowers with plastic circles, suggesting that they recognized the specific morphology of the ants. Also, they showed that pollinators spent 2.5 times more time per flower on the ant-free branches, and the number of fruits produced per flower was 2 times lower in the group with plastic ants. 

In addition to size and aggressiveness, the number of ants foraging on plants and the pheromones they release are indicators used by pollinators. Large numbers of ant individuals (workers and/or soldiers) might be seen from far distances, dissuading pollinators from visiting flowers [44]. For instance, using field observations and experimental manipulations of ant density in the branches of *Banisteriopsis malifolia* (Malpighiaceae), Barônio and Del-Claro [44] demonstrated a significant decrease in the number of visits and fruit set as the density of ants increased. Ants release trail and territorial pheromones produced by different glands, such as the metapleural glands, to communicate or dominate territories [46,47,48]. Pollinators, in turn, can learn to use these scent marks as informational signs or associate them with the risk of attack or danger, avoiding visiting plants with ants [49,50,51]. Ballantyne and Willmer [50] demonstrated in laboratory tests that bee drones learn to use ant scent marks as informational signals during foraging on artificial flowers. In another study, Cembrowski et al. [49] showed that *Bombus impatiens* bees collect less pollen from artificial flowers that ants have crawled on compared to controls, suggesting that some bees use ant scent marks to avoid conflict with ants when visiting flowers. 

Finally, the pollinator responses also vary depending on the pollinator identity. Some pollinators are more likely to visit flowers containing ants than other pollinators. For instance, in a study conducted by Barônio and Del-Claro [44], the presence of ants around the flowers of Malpighiaceae plants (e.g., Figure 1E,F) reduced the visitation rates of smaller non-Centridini bees but not larger (usually >1 cm) Centridini bees. Studying the interaction between the weaver ant *Oecophylla smaragdina* and the tropical shrub *Melastoma malabathricum*, Gonzálvez et al. [52] observed that smaller bees tended to visit ant-free plants, while large bees concentrated their visits on plants occupied by ants. Other studies have also shown this pattern, in which smaller bees tend to approach more carefully flowers containing ants [25,44,49,53], reinforcing the evidence that the identity of the floral visitors in addition to the identity of the ants is also a factor determining the trade-off between protection and deterrence. 

## 3. Positive Effects of Pollinator Repellence by Ants

Although there seems to be a clear negative effect of the presence of ants when foraging on flowers on pollinator behavior, a few studies have shown positive outcomes [23,33,51,54]. Decreasing pollinator visitation to flowers in response to the presence of ants may lead to decreased self-pollination or may increase the likelihood of cross-pollination, as pollinators are forced to constantly change flowers within plants and/or to move to other plants due to the presence of ants [23,33,51,54]. In this context, there is a decrease in the foraging efficiency of pollinators on the same plant individual but an increase in movement between individual plants, bringing positive effects for plant reproductive success. Villamil et al. [29] provided evidence of the deterrence of floral visitors by ants in an ant–plant interaction mediated by EFNs. Studying the shrub *Turnera velutina* (Passifloraceae), they showed that the more aggressive the ant, the more likely it is to deter floral visitors; however, they also showed that this deterrence reduced three-fold self-pollination and increased cross-pollination two-fold (also see [51]). We are just beginning to understand what the net outcomes of this trade-off between protection from herbivore attack and pollinator deterrence are, as well as which biotic factors may affect the outcomes of these interactions. By addressing these questions, we can elucidate the costs and benefits, and therefore the net outcomes, provided by ant mutualists to plants bearing EFNs.

## 4. Plant Traits to Reduce Ant Impacts on Pollination Processes

Assuming that there is a trade-off between the benefits of protection against attack by herbivores and the disruption of pollination processes by ants, there are ecological and evolutionary aspects inherent to these interactions that can balance this trade-off (Box 1, Table 1). There is some evidence that plants have several mechanisms to lessen the impacts caused by ants on plant reproductive success [30,55,56,57,58]. Some of these mechanisms have already been extensively tested, while others are based only on a few investigations. For example, plants have evolved floral and foliar volatile organic compounds repellent to ants; narrow tubes of nectar, limiting ant access to floral rewards to pollinators; repellent nectar and pollen; trichomes close to the reproductive parts, making it difficult for ants to access the reproductive parts; sticky or greasy structures around the reproductive parts; and EFNs near to or associated with the reproductive parts, which can lure ants to forage outside the flower, decreasing the probability of encounters with pollinators (Box 1, Table 1), known as the flower distraction hypothesis [32]. The presence of EFNs near to or associated with the reproductive parts could, to some degree, attract ants, which protect the plant from the attack of florivores, [59,60] and/or distract ants so that they do not reach the flower buds and flowers (Figure 1E–G), resulting in potential negative effects on the pollinator behavior and plant fitness [30,32,61]. The dynamics of extrafloral nectar production (constitutive or induced) and the quantity and quality of nectar produced are important factors for regulating ant attendance and aggressiveness [41,60,62,63,64,65,66,67,68,69]. Future studies should evaluate each EFN-bearing plant as an isolated system, mesuaring the herbivory rates and the impacts of ants on pollinator deterrence when manipulating the presence of ants, pollinators, florivores, and EFNs. 

Box 1Evolutionary plant mechanisms to deter ants from visiting reproductive parts.  Ants are often poor pollinators [47,70] and can kill pollen with their glandular secretions [71], deter other pollinators [26,27], and even collect pollen and nectar without effecting pollination [21] (Table 1). Thus, the results of different studies have suggested that ant repellence from flowers might increase plant reproductive success [57,72,73,74]. To lessen these potential negative impacts of ants when they visit flowers, certain plants release floral and foliar volatile organic compounds, which are used as chemical signals to control ant behavior, attracting them to the leaves and/or preventing them from accessing the reproductive parts [56,57,73,74] (Table 1). Floral nectar can be a rich source of carbohydrates for ants in the absence of EFNs, as the majority of plants in most habitats do not bear EFNs [58,75]. The quality and palatability of the floral nectar are just some of the factors that can repel ants [58]. Due to secondary metabolites being dissolved in the sugar solution, the floral nectar can become unpleasant or toxic, resulting in rejection by ants [76,77,78,79]. This type of nectar, therefore, has the function of repelling the visitation of floral ants [79,80,81] (Table 1). In addition to chemical compounds, plants can also use mechanical or morphological barriers to decrease ant foraging on the reproductive parts (Table 1). For instance, some plants use trichomes as a potential mechanism to hinder or decrease foraging of ants and bocking their access to the reproductive parts [58,82]. Junker et al. [58] found very sticky glandular hairs in the calyx of *Plumbago zeylanica* (Plumbaginaceae), which act as a barrier to crawling insects, and fine and dense hairs that cover the calyx of *Abutilon eremitopetalum* (Malvaceae), preventing ants from reaching the floral nectaries. Another type of morphological barrier is related to nectar accessibility, in which narrow floral tubes prevent ants from accessing nectar [83,84,85]. 

**Table 1 plants-13-00651-t001:** Plant traits that can decrease the negative effects of ants against pollinators.

**Mechanism**	**Acting**	**References**
Floral and foliar volatile organic compounds	Plants release floral and/or foliar volatile organic compounds to control the behavior of ants, repelling them from flowers and/or attracting them to the leaves	Floral—[57,74] Foliar—[86,87]
EFNs in reproductive parts	Plants offer extrafloral nectar close to their reproductive parts, serving as a distraction to prevent ants from reaching the flower buds and flowers—flower distraction hypothesis	[30,31,32,61,88]
Extrafloral nectar quantity and quality	Variation in the extrafloral nectar volume and sugar concentration can change ant aggressiveness, which can potentially decrease pollination disruption	[62,63,64,67]
Dynamics of production of extrafloral nectar	The production of extrafloral nectar in the reproductive parts can be induced rather than constitutive, attracting ants only after attack or risk of attack	[60,65,89]
Narrow tubes of nectar	Narrow tubes of nectar, as well as flowers with short, narrow corollas, prevent ants from reaching the nectar	[58,85]
Repellent nectar	Floral nectar that is toxic or repellent to ants	[58,79]
Repellent pollen	Toxic or repellent pollen to ants	[56,90]
Trichomes	The presence of trichomes can hinder or decrease the access of foraging ants to flowers	[55,58,82]
Special features	Sticky or greasy structures outside the corolla or on the flower pedicel	[91]

## 5. Optimal Ant Mutualist Partners

Considering the existence of a trade-off between the protection of plants and the disruption of pollination processes, we suggest the existence of an “optimal ant mutualist partner”, equivalent to effective pollinators in many pollination systems. The flowers of pollinator-dependent plants are often visited by different floral visitors, in which not all visitors play pollinator roles; see a review in [34]. Those who visit flowers frequently and touch the reproductive parts are considered effective pollinators, while others which do not visit the flowers frequently but touch the reproductive parts are considered occasional pollinators [34]. Comparing the protection mutualisms between ants and plants mediated by EFNs and the pollination system, aggressive ants that significantly protect plants compared to non-aggressive ants and cause less impacts on pollination processes than other aggressive ants could be considered optimal (effective) mutualist partners. This optimal ant partner therefore represents the optimum point of the balance between positive effects on plant protection and negative effects on the pollination process (Figure 3).

In general, we propose a conceptual framework for understanding the impacts of different levels of ant aggressiveness on factors related to the protection and reproduction of plants; see [28,92]. Docile ants (*point a* in Figure 3) with low levels of aggressiveness tend to show a low effect on protection effectiveness (Figure 3A), pollinator deterrence (*point a* in Figure 3B), and the probability of cross-pollination (Figure 3D) but high effects on self-pollination probability (Figure 3C) and pollen removal/deposition (Figure 3E). Ants with high levels of aggressiveness (*point b* in Figure 3) tend to show a high effect on protection effectiveness (Figure 3A), pollinator deterrence (Figure 3B), and cross-pollination (Figure 3D) and low effects on self-pollination probability (Figure 3C) and pollen removal/deposition (Figure 3E). Finally, some ants (potential optimal partners; *point c* in Figure 3) are aggressive enough to protect plants (Figure 3A) with a lower impact on pollinator deterrence than very aggressive ants (Figure 3B), decrease the probability of self-pollination (Figure 3C), and increase the cross-pollination probability (Figure 3D) while allowing enough pollen deposition/removal (Figure 3E). To corroborate the optimal protective mutualist hypothesis, studies should evaluate, when possible, these different effects (protection and pollination disruption) on plants caused by different extrafloral nectar-feeding ants in each system (Box 2), while also quantifying when possible impacts on plant fitness (e.g., fruit and seed set).

Box 2Evidence for optimal protective ants.  This existence of an optimal ant partner seems to be common in the Brazilian savanna. In this region, 25% of woody plant species and 30% of individuals at a site were found to have EFNs [93,94,95]. The main group of numerically and behaviorally dominant ants foraging on these plants is *Camponotus* [60,92,94,96,97,98], with great emphasis on *C. crassus*, which has been recorded as the most abundant and protective ant on EFN-bearing plants in different areas of the Brazilian savanna [2,63,64,68,92,94,96,97,99]. For instance, Souza et al. [92] showed that *Camponotus* ants are the best EFN-bearing plant protectors according to baiting tests. Studying *Qualea multiflora* plants, we tested how ants’ presence on flowers can impact pollinator deterrence and plant fitness (Calixto et al. in review) and whether this impact is contingent on ant and floral visitor identity. We used three different ant species based on a combination of aggressiveness and size [21,25,28]: *Cephalotes pusillus*, smaller and non-aggressive; *Camponotus crassus* (Figure 1I), medium-size and aggressive; and *Ectatomma tuberculatum* (Figure 1H), larger and very aggressive. We observed that the largest and most aggressive ant, *E. tuberculatum* (Figure 1H), had the greatest negative impact on pollinator deterrence and plant reproduction, *C. pusillus* had no impact, and *C. crassus* was weakly negative. Byk and Del-Claro [21] showed that *C. pusillus* does not protect plants against foliar herbivory and also consumes pollen. The other two ants are effective at protecting EFN-bearing plants [63,64,97,100], but *E. tuberculatum* has been associated with negative impacts on pollinators and plant fitness; also see [25,28]. Finally, in addition to protecting plants, *C. crassus* has been shown to have a low impact on pollination and does not possess metapleural glands [48] as *E. tuberculatum* ants do, which can make pollen unviable. Supported by these studies, we suggest that *C. crassus* is a species aggressive enough to protect plants against herbivore attack but leads to low negative impacts on pollination processes (*point c* in Figure 3). Therefore, *Camponotus* ants, specifically *C. crassus*, might represent the suggested “optimal protective mutualist” (Figure 3) in some areas of the Brazilian savanna where they are dominant on EFN-bearing plants; also see [28,92]. 

## 6. Concluding Remarks and Future Directions

Our study reviewed the existence of a trade-off between the benefits generated by the protection against the attack of herbivores and the losses caused by the disruption of pollination processes, in which the balance of this trade-off is contingent on the different traits of the organisms involved. Knowing the identity of the ants, herbivores, plants, and pollinators can be essential to understanding the levels of protection and pollination disruption established. More aggressive ants are expected to provide better protection to plants against herbivore attack, but at the same time they can have a greater influence on pollination processes than less aggressive ants, showing context-dependent outcomes [54,101,102,103,104]. 

Indeed, there is an ant–pollinator conflict, and this conflict can often bring negative outcomes to plants. However, plants have evolved different mechanisms to decrease the disruption of pollination processes (Table 1). Our understanding of which mechanisms are acting in any given ant–plant–pollinator system is often unknown. We believe that, in general, the presence of EFNs decreases the herbivory rate and in some cases increases plant reproductive success [19,20], but in many other systems, we do not know the conditional effects of these ants on plant fitness, which can be negative, neutral, or even positive. Thus, we suggest that each system mediated by EFNs should be evaluated individually to better understand the magnitude (strength) and direction (positive, negative, or neutral) of the effects of the associated ants.

An important scenario that should also be evaluated is assessing the influence of ant aggressiveness and consequently the protection effectiveness according to the quantity and quality of and the dynamics of production (constitutive or induced) of extrafloral nectar [2,62,63,67]. Some studies have shown the important role of the production of extrafloral nectar on ant foraging behavior, which can influence their number, aggressiveness, and protection effectiveness [60,62,66,96,105]. For instance, Calixto et al. [62] showed significant variation in the protective effectiveness of *Camponotus crassus* for five sympatric EFN-bearing plants in the Brazilian savanna. Although not evaluated in their study, this variation in the levels of aggressiveness could reflect variable impacts on pollination processes. In addition, the production of extrafloral nectar in the reproductive parts can be induced rather than constitutive, attracting ants only after attack or risk of attack [60]. Thus, the abundance of ants close to the reproductive parts would be low, potentially decreasing their impacts on pollination processes. Therefore, bottom-up control from plants on ant foraging behavior [62,63,64,65,67] might determine the balance of the potential trade-off between protection against herbivores and the deterrence of pollinators. 

Finally, it would be important to assess which plant traits are most likely to be prone to the negative effects of ants on pollinators. For instance, EFN-bearing plants that exhibit different growth forms (e.g., shrubs versus trees) may exhibit different ant impacts. Shrubs might be more likely to have their flowers visited by ants due to their smaller size compared to trees. EFNs located in the reproductive regions may attract more ants to the flowers than EFNs in the vegetative regions, increasing their likelihood of impacting pollination processes. Plants that flower while EFNs are active may also be more likely to have their pollination processes impacted by ants. Finally, flowers with resources accessible to ants, for example, exposed nectar or pollen, may be more affected than plants that present nectar and pollen in specific structures (e.g., spurs and poricidal anthers). 

As discussed in this review, EFNs play a multifaceted role in mediating ant–plant interactions. While they are often associated with pollination processes, their significance extends beyond this realm [23,106,107]. In the context of post-pollination aspects, the dynamics of EFNs become particularly important [51]. For instance, the optimal ant protective mutualist can confer benefits to plant fitness by causing little or no disruption to pollination processes. However, this mutualistic association may falter when it comes to fruit and seed protection, potentially jeopardizing all the previous advantages in terms of plant fitness [106]. In such cases, the benefits of the mutualistic relationship may shift from pre-pollination to post-pollination processes [51]. In this context, we suggest that, to better classify ants as optimal protective mutualists in their systems, we should evaluate ants’ effects on plant fitness across various pollination processes, ranging from pre-pollination to the post-pollination stages [51,106,107].

Supported by many studies related to ant–plant mutualisms, we point out that protective mutualisms can also have optimal partners, as suggested for pollination mutualisms. This optimal protective mutualist would be characterized by presenting the best balance between the positive effects of protecting plants from herbivores and the negative effects of affecting processes related to pollination. We suggest that studies carry out manipulative experiments in systems under different contexts, looking for the costs and benefits of mutualist ants to plants [108], paying close attention not only to the impacts on herbivory but also on pollinators. 

## Figures and Tables

**Figure 1 plants-13-00651-f001:**
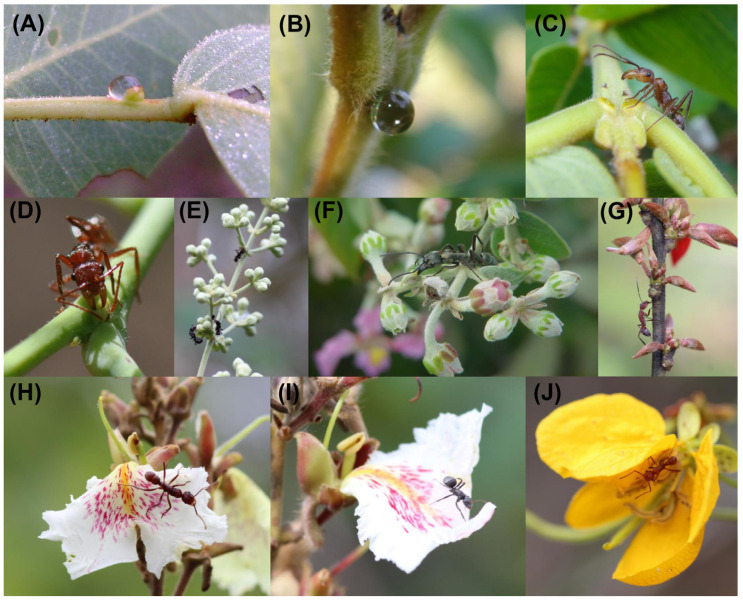
Extrafloral nectar drops in (**A**) *Stryphnodendron adstringens* (Fabaceae) and (**B**) *Qualea multiflora* (Vochysiaceae). (**C**,**D**) *Ectatomma tuberculatum* collecting extrafloral nectar. (**E**) *Ectatomma* sp. and (**F**) *Neoponera villosa* ants collecting nectar from the extrafloral nectaries on the inflorescence of *Banisteriopsis malifolia* (Malpighiaceae). (**G**) *Camponotus leydigi* collecting nectar from the extrafloral nectaries on the inflorescence of *Bionia coriacea* (Fabaceae). (**H**) *Ectatomma tuberculatum* and (**I**) *Camponotus crassus* foraging on the flowers of *Q. multiflora*. (**J**) *Ectatomma tuberculatum* foraging on the flowers of *Senna rugosa* (Fabaceae). Photos: E.S. Calixto, D. Lange, H. M. Torezan-Silingardi, K. Del-Claro.

**Figure 2 plants-13-00651-f002:**
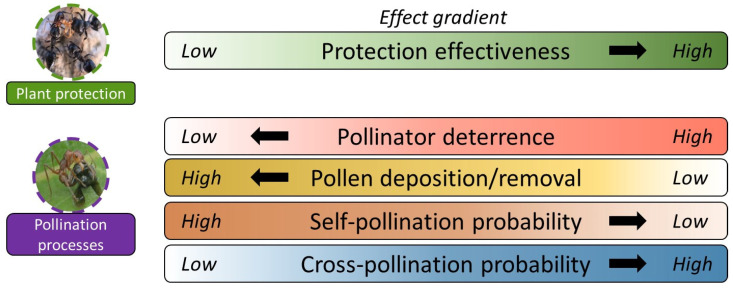
Trade-off between plant protection and disruption of pollination processes. Hypothetical framework for understanding the potential trade-off between plant protection against herbivore attack and negative effects on pollination processes by mutualistic ants attracted by extrafloral nectaries. Ants protect plants against herbivore attacks (positive effect), but due to their different levels of protection effectiveness and other traits (e.g., size, pheromones), they may impair the pollination process, ultimately resulting in a decrease in plant fitness (negative effect). An ant species that has the highest positive effects in protection effectiveness and the lowest negative effects on pollination processes is therefore an optimal mutualist partner (black arrows show the direction of the effects of optimal ant partners on plant protection and pollination processes).

**Figure 3 plants-13-00651-f003:**
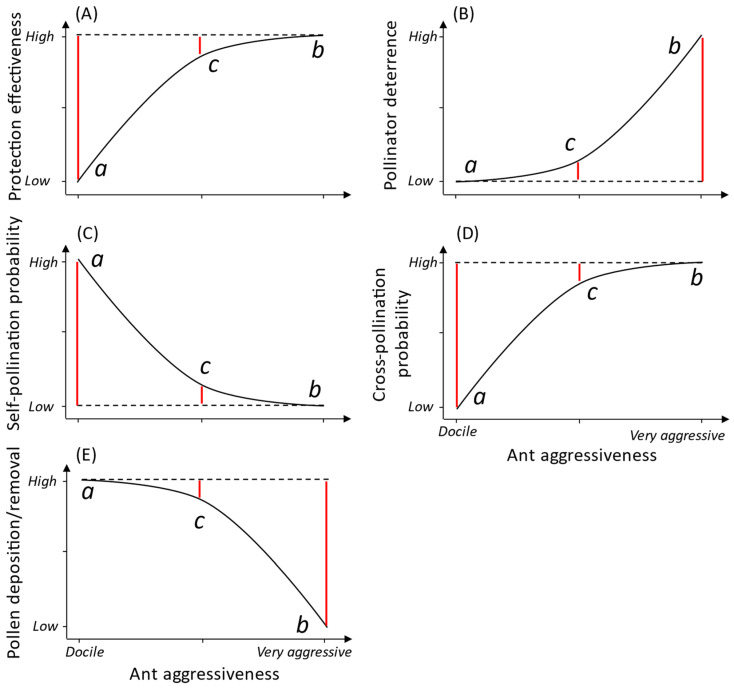
Association between ant aggressiveness and the positive and negative impacts on EFN-bearing plants. Potential outcomes of ant aggressiveness (or identity) on plant protection effectiveness (**A**) and pollination processes, such as pollinator deterrence (**B**), self-pollination probability (**C**), cross-pollination probability (**D**), and pollen deposition/removal (**E**) in ant–plant–pollinator interactions mediated by extrafloral nectaries. *Point a* represents a docile ant (usually small and non-aggressive compared to other ants in the same system), *point b* represents a very aggressive ant (compared to other ants in the same system, it is usually large and very aggressive toward herbivores and pollinators), and *point c* represents an aggressive ant (compared to other ants in the same system, it is usually medium-size and aggressive toward herbivores but not pollinators). *Point c* represents the optimal balance between ant aggressiveness and the impacts on plant protection and pollination processes. Non-linear solid black lines represent the relationship between ant aggressiveness and plant protection effectiveness and pollination processes. Dashed lines represent the best scenario for plants. Vertical solid red lines represent the magnitude of the negative effects caused by ants; the smaller, the better.

## Data Availability

No new data were created or analyzed in this study. Data sharing is not applicable to this article.

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
