# Peer review of "Emerging Trends in Ant–Pollinator Conflict in Extrafloral Nectary-Bearing Plants"

_plants, 2024, doi:10.3390/plants13050651_

Round 1

Reviewer 1 Report

Comments and Suggestions for Authors

The manuscript submitted for the assessment contains the review of 97 publications on the ecological functions of ants visiting plants producing extrafloral nectaries. Publications from the last 10 years constitute 43% of the analysed literature.

The topic of the review is interesting from the point of view of plant protection by ants against herbivores and the ecology of pollination and reproductive success of plants. The paper is an attempt to explain the functioning of the ant-plant-pollinator system in various plant species and presents the positive and negative effects of these relationships.

The manuscript seems to be missing a few photographs related to the topic.

In my opinion, the paper can be published in its current version , possibly supplemented with relevant photographs.

Author Response

We thank the Associate Editor and reviewers for their constructive criticisms and are pleased to return to you a revised and improved version of our manuscript (plants-2884301) entitled “Emerging trends in ant-pollinator conflict in extrafloral nectaried plants”.

We added a new figure in the manuscript as recommended by Reviewer #1, and we addressed the two minor points suggested by Reviewer #2.

We hope that this version will satisfy your concerns. We look forward to your response.

Kleber Del-Claro

Reviewer 2 Report

Comments and Suggestions for Authors

This is a thorough review of issues pertaining to trade-offs in benefits from ant -plant mutualisms associated with extrafloral nectaries and risks associated with reduced pollination associated with the presence of ants. The organization and argumentation are both sound, and the language is clear and easy to read. I have no substantive suggestions for improvement, and I think the manuscript is ready to go essentially as is.

I did find two minor points that need fixing. First, lines 87-89 need to be deleted as these were left over from the manuscript preparation template. Second, the abbreviation FDH in line 191 (and in Table 1) is undefined in the text.

Author Response

(The authors gave the same response as above.)

Reviewer 3 Report

Comments and Suggestions for Authors

The authors realised a nice review about emerging trends in the ant-pollinator conflict in extrafloral nectaried plants. They summarised current knowledge, drew tendencies, and set up questions based on currently available literature that should be assessed in the coming research.

Author Response

We thank the Associate Editor and the three reviewers for their constructive criticisms and are pleased to return to you a revised and improved version of our manuscript (plants-2884301) entitled “Emerging trends in ant-pollinator conflict in extrafloral nectaried plants”.

We added a new figure in the manuscript as recommended by Reviewer #1, and we addressed the two minor points suggested by Reviewer #2.

We hope that this version will satisfy your concerns. We look forward to your response.

Kleber Del-Claro
